# Nursing Student Satisfaction Scale: Evaluation of Measurement Properties in Nursing Degree Programs

**DOI:** 10.3390/nursrep15050161

**Published:** 2025-05-03

**Authors:** Rocco Mazzotta, Giampiera Bulfone, Bartolomeo Verduci, Vera Gregoli, Davide Bove, Massimo Maurici, Ercole Vellone, Rosaria Alvaro, Francesco Scerbo, Maddalena De Maria

**Affiliations:** 1Department of Biomedicine and Prevention, University of Rome Tor Vergata, 00133 Rome, Italy; rocco.mazzotta@uniroma2.it (R.M.); bartolomeo.verduci@alumni.uniroma2.eu (B.V.); vera.gregoli@students.uniroma2.eu (V.G.); davide.bove@students.uniroma2.eu (D.B.); maurici@med.uniroma2.it (M.M.); ercole.vellone@uniroma2.it (E.V.); rosaria.alvaro@uniroma2.it (R.A.); 2Department of Medical, Surgical Sciences and Advanced Technologies, University of Catania, 95123 Catania, Italy; giampiera.bulfone@unict.it; 3Mediterranean Centre of Excellence for the Academic Development of Nursing Research, 80134 Naples, Italy; 4Faculty of Nursing and Midwifery, Wroclaw Medical University, 50-996 Wrocław, Poland; 5Center of Excellence for Nursing Culture and Research, Order of Nursing Professions of Rome, 00136 Rome, Italy; 6Department of Life Science, Health, and Health Professions, Link Campus University, 00165 Rome, Italy; m.demaria@unilink.it

**Keywords:** satisfaction, nursing student, nursing education, validity, reliability, Nursing Student Satisfaction Scale (NSSS)

## Abstract

**Background/Objectives**: Satisfaction among undergraduate nursing students plays a crucial role in student retention, helping to mitigate nursing shortages in the workforce, reduce academic costs, and uphold universities’ reputations. The Nursing Student Satisfaction Scale (NSSS) measures three theoretical domains: Professional Social Interaction, Curriculum and Teaching, and Learning Environment. Although the NSSS has demonstrated reliability and validity with respect to American nursing students, its psychometric properties have not been tested on a population of Italian nursing students. Therefore, this study aimed to determine the reliability and validity of the NSSS in regard to Italian nursing students. **Methods**: A multicenter observational study was conducted on undergraduate nursing students in Central Italy. A confirmatory approach was used to assess structural validity. The construct validity, internal consistency, test–retest reliability, and responsiveness to change of the NSSS were evaluated using correlation analyses, reliability coefficients, and measurement error determination. **Results**: Confirmatory factor analysis supported the three-factor first-order structure of the NSSS as well as the presence of a single second-order factor. Reliability was adequate for all the coefficients computed (with values ranging from 0.924 to 0.974). Construct validity was supported. The measurement error was adequate. **Conclusions:** The NSSS exhibited robust measurement properties, confirming its validity and reliability as an instrument for assessing nursing student satisfaction in the Italian context. Furthermore, our results indicate that, after the translation and cultural adaptation of the scale, the construct of nursing student satisfaction remains consistent with the theoretical model.

## 1. Introduction

Satisfaction within an academic context is a complex, multidimensional [1,2,3], and dynamic construct [4,5]. Student satisfaction can be defined as a subjective assessment made by a student and determined by the gap between the student’s expectations and their perception of their educational experiences [6,7]. According to Chen et al. [4], student satisfaction is the result of a dynamic process influenced by the interaction between students, faculty, and the educational environment, structured around four core dimensions: curriculum, faculty, social interaction, and learning environment.

Student satisfaction is a recognized indicator of program quality in higher education [8] and a key component in monitoring and continuously improving educational quality, as outlined by the European Standards and Guidelines for Quality Assurance in the European Higher Education Area (ESG) [9]. Universities that engage in continuous quality assurance activities should meet students’ expectations and needs [10], as these aspects significantly influence student retention rates, a critical challenge, particularly in nursing degree programs due to the high academic demands, emotional burden, and the complexity of clinical training, which may lead to increased stress levels and withdrawal intentions [11,12]. Several studies have highlighted that student satisfaction is associated with factors that promote retention in nursing education, including perceived institutional and peer support, the presence of effective and stimulating learning environments, and high-quality clinical experiences [13,14,15]. Satisfied students are more likely to achieve both academic and social integration, which have been identified as protective factors with respect to program attrition. Such integration not only contributes to individual success but may also help mitigate the broader issue of nursing workforce shortages, a growing concern at the national and international level [16,17]. Furthermore, program attrition entails significant costs for educational institutions, leading to suboptimal utilization of available resources and potentially damaging the reputation of nursing programs, especially since high dropout rates are perceived as indicators of inadequate educational quality [13]. On the contrary, nursing student satisfaction contributes to enhancing students’ self-confidence and self-efficacy, which are essential for the development of knowledge, clinical competencies, and academic performance [18,19]. Additionally, satisfaction is associated with increased motivation and engagement, factors widely recognized as fundamental drivers of effective learning processes and academic achievement [20].

Currently, the available instruments for measuring nursing student satisfaction often lack alignment with theoretical reference models and fail to follow a rigorous methodology for their development and validation [21]. A systematic review conducted by Rossini et al. [22] has identified five instruments used to assess nursing student satisfaction in nursing education. However, the validation processes of these instruments rarely involved the employment of a confirmatory approach to assess structural validity, and none of the studies examined measurement error or criterion validity. Despite these methodological limitations, Rossini et al. [22] concluded that the best available instrument for measuring nursing student satisfaction was the Nursing Student Satisfaction Scale (NSSS), originally developed in 2012 by Chen et al. [4], due to its solid theoretical model and the rigor of its development process.

The NSSS was developed and tested using 303 American nursing students [4]. Its construct validity was tested using Principal Component Analysis (PCA), while its internal consistency reliability was evaluated through Cronbach’s alpha. The results showed a three-factor factorial structure: Professional Social Interaction (which involves interpersonal relationships, interaction, respect, trust, and active participation in the teaching–learning process in educational settings), Curriculum and Teaching (regarding faculty qualifications, curriculum structure and coherence, teaching methodologies, and the alignment between theory and practice), and Environment (which concerns the teaching resources and facilities, along with the procurement and upkeep of equipment). An adequate internal consistency reliability (a Cronbach’s alpha ranging from 0.86 to 0.93 for the three factors) was shown [4].

In addition, Hirsch and colleagues [23] tested the NSSS on a sample of 123 Brazil undergraduate nursing students. Its construct validity was tested by using exploratory factorial analysis (EFA), while internal consistency reliability was tested by using Cronbach’s alpha coefficient. The EFA results showed a three-factor factorial structure (curriculum and teaching; professional social interaction; and learning environment factors) after eight items were deleted. One item (“Program prepared me to take the NCLEX-RN”) was reformulated to culturally adapt it to the context of Brazilian students. The Cronbach’s alpha ranged from 0.88 to 0.93.

To the best of our knowledge, no previous study has evaluated the construct validity of the NSSS for a sample of undergraduate nursing students by using a confirmatory statistical approach, which is recommended for testing hypothesized factorial structures and validating theoretically based measurement models. Unlike exploratory methods such as Exploratory Factor Analysis (EFA) or Principal Component Analysis (PCA), confirmatory approaches allow for theory-driven validation by testing pre-specified factor structures according to the theories, assessing model fit through standardized indices, and generating robust and replicable evidence of construct validity across different samples [24]. Furthermore, no prior research has investigated the internal consistency reliability of the NSSS while accounting for its multidimensional structure. Estimating reliability using coefficients specifically designed for multidimensional scales (i.e., those that incorporate factor loadings and inter-factor correlations) yields more precise and theoretically appropriate estimates than the traditional unidimensional indices previously tested.

In this context, measuring nursing student satisfaction through reliable and valid instruments becomes crucial not only for monitoring educational quality but also for informing targeted interventions aimed at improving learning environments, reducing attrition, and ultimately contributing to the preparation of competent nursing professionals capable of responding to current and future healthcare challenges.

Therefore, in this study, we aimed to test the psychometric characteristics (structural validity, construct validity, and internal consistency reliability) of the NSSS using a sample of undergraduate nursing students.

## 2. Materials and Methods

### 2.1. Design

A multicenter observational study was performed. This study is reported in compliance with the EQUATOR guidelines, specifically following the Strengthening the Reporting of Observational Studies in Epidemiology (STROBE) guidelines for cross-sectional studies [25].

### 2.2. Participants and Setting

A convenience sample of undergraduate nursing students was recruited from multiple educational sites offering nursing degree programs in Central Italy. Eligible participants were those enrolled regularly in the program, with no restrictions based on the year of enrollment, and those had provided informed consent to participate in this study. The students who had expressed an intention to change their field of study were excluded.

### 2.3. Data Collection

Data collection was carried out during scheduled classroom sessions at the end of the second semester of the academic year, allowing students to provide a more comprehensive and informed evaluation of their learning experience; each session lasted an average of 30 min and was held between April and May 2024. Data collection was carried out by research assistants who were all PhD students in nursing. These assistants were not involved in the educational activities of the participants and were adequately trained by the research team in standardized data collection procedures. During the administration of the questionnaire, the research assistants were present and available to respond to any questions or clarify potential doubts raised by the participants, thereby ensuring consistency and accuracy in data collection.

### 2.4. Measurements

The NSSS [4] is a self-report measure designed to evaluate nursing student satisfaction. It consists of 30 items scored on a 6-point Likert-type scale (ranging from 1, “not satisfied”, to 6, “very satisfied”) grouped into three factors that measure “Professional Social Interaction” (9 items), “Curriculum and Teaching” (13 items), and “Environment” (7 items). Cronbach’s alphas were 0.91, 0.90, and 0.86, for Professional Social Interaction, Curriculum and Teaching, and Environment, respectively, while the value was 0.93 for the overall scale.

The Italian version was translated from the original English version in accordance with the Principles of Good Practice for the Translation and Cultural Adaptation Process for Patient-Reported Outcomes (PRO) measures [26]. The process commenced with two bilingual translators independently translating the instrument from English to Italian. A third translator, in collaboration with the research team, reviewed and harmonized the translations to ensure consistency and accuracy. Subsequently, two additional translators independently back-translated the reconciled Italian version into English to verify its accuracy. The research team then carefully reviewed and refined the back-translated version, resolving any ambiguities in collaboration with the translators. According to a previous study [23], we used a 5-point Likert scale ranging from 1 (“very dissatisfied”) to 5 (“very satisfied”), where higher scores indicate higher nursing student satisfaction.

The Academic Nurse Self-Efficacy scale (ANSEs) [27] is a self-report instrument used to evaluate the academic self-efficacy of nursing students. It consists of 14 items grouped into four factors: auto-regulatory behavior (4 items), external emotion management (4 items), collegiality (3 items), and internal emotion management (3 items). The scale has good reliability, with Cronbach’s alpha coefficients of 0.72, 0.83, 0.73, and 0.80 for external emotion management, collegiality, and internal emotion management, respectively [27]. The total scale has a Cronbach’s alpha of 0.84. In this study, the ANSEs was used to test the construct validity of the NSSS.

Socio-demographic variables such as age, gender, year of enrollment, marital status, number of children, housing situation, employment status, and whether a nursing degree was the participant’s first choice were collected using a questionnaire specifically developed by the research team to characterize the sample and provide context for the interpretation of the factorial analysis.

### 2.5. Ethical Considerations

This study was carried out in accordance with the principles outlined in the Declaration of Helsinki and in compliance with current clinical trial regulations. The research protocol received approval from the Ethics Committee (No. 254/22). All potential participants were thoroughly informed about the study’s objectives and assured of the confidentiality of their data. Written informed consent was obtained from all participants prior to their involvement in the study.

### 2.6. Data Analysis

Descriptive statistics, including frequency, percentage, mean, and standard deviation (SD) coefficients where appropriate, were calculated to characterize the sample and the NSSS items. The normality of the NSSS items was assessed by evaluating kurtosis and skewness [28]. In line with the recent literature, the analysis began with an assessment of the scale’s structural validity, followed by an evaluation of its reliability [29].

A confirmatory approach was used to test the structural validity of the scale. Specifically, a confirmatory factor analysis (CFA) was conducted. Due to the non-normal distribution of the NSSS items, the maximum likelihood robust (ML-r) estimator was used for parameter estimation [30]. According to the structural model proposed by Chen during the development of the NSSS, a three-factor model (“Professional and Social Interaction”, “Curriculum and Teaching”, and “Environment”) was tested. Additionally, considering the correlations between factors, a second-order model was hypothesized. Model fit was examined by using the following fit indices: comparative fit index (CFI), Tucker and Lewis index (TLI), root mean square error of approximation (RMSEA), and standardized root mean square residual (SRMR) [31,32]. The chi-square statistics were also considered together with the indices above. The goodness-of-fit values were interpreted in accordance with recommendations from the existing literature [33,34].

The construct validity of the NSSS was assessed through hypothesis testing by examining the correlations among the scales. Pearson’s correlation coefficient was employed to test the hypotheses. Correlation coefficients were interpreted as follows: values between 0.10 and 0.29 indicated weak correlations, values between 0.30 and 0.50 indicated moderate correlations, and values greater than 0.50 indicated strong correlations [35]. Several hypotheses were posed: (i) the scores of each factor comprising the NSSS will be positively correlated; (ii) there will be positive correlations between the scores of ANSEs and each NSSS factor. The rationale underlying these hypotheses is that self-efficacy contributes to making a student’s overall educational experience more satisfying by strengthening the key variables that influence the learning process [36].

The internal consistency reliability of the NSSS was assessed by calculating the composite reliability coefficient for each factor [37]. The global reliability index for multidimensional scales [38], a more appropriate reliability coefficient that considers the scale’s multidimensionality, was also computed to test the reliability of the overall NSSS. Values ≥ 0.70 were considered adequate [39].

To assess the stability of the NSSS, test–retest reliability was evaluated by administering the instrument to a subset of nursing students over a two-week interval. Intraclass correlation coefficients (ICCs) were calculated for each scale using a two-way random-effects model. An ICC value of ≥0.75 was interpreted as indicating good reliability, while a value of ≥0.90 was considered indicative of excellent reliability [40].

To assess responsiveness to change, the precision of the NSSS instrument was evaluated by calculating the measurement error, specifically the standard error of measurement (SEM) and the smallest detectable change (SDC). The SEM was determined using the following formula: standard deviation (SD) × √(1 − reliability coefficient) [41], where the SD refers to the NSSS score, and the reliability coefficient corresponds to the composite reliability. Lower values of SEM and SDC indicate greater precision of the instrument.

Statistical analyses were conducted using SPSS Version 26, with the exception of the CFA, which was performed using Mplus Version 8.4 [42].

## 3. Results

### 3.1. Characteristics of the Sample

Table 1 presents the characteristics of the total sample (N = 724). In brief, the students had a mean age of 23.6 (±5.7) years, were mostly female (74.0%), were enrolled in the first year of the nursing program (36.2%), were single (82.2%), did not have children (95.0%), lived with other students (56.1%), were unemployed (87.3%), and reported that nursing was their first choice (68.0%).

### 3.2. Descriptive Analysis of the Items of the Nursing Student Satisfaction Scale (Italian Version)

Table 2 presents the characteristics of the sample and the items of the NSSS. The mean score of each item ranged from 2.64 to 3.91. The lowest mean scores (indicating lower satisfaction) were found for the following items: #16 (2.64, SD 1.22), #12 (2.80, SD 1.25), #20 (2.89, SD 1.25), and #7 (2.87, SD 1.25). The highest means scores (indicating higher satisfaction) were found for the following items: #23 (3.91, SD 0.86), #9 (3.66, SD 0.97), #2 (3.64, SD 1.00), and #25 (3.60, SD 0.95). Three items had kurtosis indices > |1|, showing the non-normality of item distribution.

### 3.3. Testing the Structural Validity of the Nursing Student Satisfaction Scale (Italian Version)

According to the structural model identified in previous studies conducted by Chen et al. [4], a three-factor model was tested, namely, Professional Social Interaction (#3, #21, #29, #13, #2, #11, #7#, #1, and #5), Curriculum and Teaching (#18, #23, #8, #27, #19, #14, #22, #28, #30, #10, #9, #25, #6, and #4), and Environment (#7, #15, #20, #24, #26, #16, and #12).

The goodness-of-fit indices of this model were adequate, as shown by the following metrics: χ^2^ (401, N = 724) = 1112.600, *p* < 0.001, CFI = 0.940, TLI = 0.935, RMSEA = 0.042 (90% CI = 0.046–0.053, *p* < 0.605), and SRMR = 0.042. An inspection of the modification indices reveals excessive residual covariance among the following pairs of items: #15 and #7; #18 and #17; #26 and #12; and #28 and #9. We specified these covariances, and the fit model improved considerably: χ^2^ (398, N = 724) = 976.823, *p* < 0.001, CFI = 0.951, TLI = 0.946, RMSEA = 0.045 (90% CI = 0.041–0.048, *p* < 0.992), and SRMR = 0.042. All factor loadings were significant and greater than 0.480. Given the significant correlations among the three factors (ranging from 0.572 to 0.902, *p* < 0.01), a second-order model was specified, resulting in fit indices comparable to those of the first-order model. This finding suggests that the NSSS exhibits a three-factor structure at the first-order level while demonstrating one-dimensionality at the second-order level (Figure 1).

#### Construct Validity

Construct validity was assessed by examining the patterns of correlations among the three factors of the NSSS and between each factor and the total NSSS score. As expected, the scores of the three NSSS factors presented statistically significant and positive correlations between themselves and the NSSS total score; the correlations ranged from 0.572 to 0.969, with *p* < 0.01. These results support the internal structure of the instrument, confirming that the three dimensions are related but distinct components of the overarching construct measured by the NSSS.

Moreover, as theoretically expected, the NSSS total score exhibited statistically significant positive correlations with the ANSEs scores, with coefficients ranging from 0.217 to 0.355 (*p* < 0.01) (Table 3). This result suggests that higher levels of perceived stress among nursing students are associated with higher levels of self-reported academic self-efficacy among undergraduate nursing students.

### 3.4. Reliability of the Nursing Student Satisfaction Scale (Italian Version)

#### Internal Consistency Reliability and Stability

The internal consistency reliability values computed with the composite reliability coefficients of the three NSSS factors were all adequate and equal to 0.924, 0.951, and 0.926 for the Professional and Social Interaction, Curriculum and Teaching, and Environment factors, respectively (Table 4). The global reliability index for multidimensional scales was adequate and equal to 0.974. The reliability of the NSSS was evaluated through test–retest analysis by re-administering the instrument to a subsample of 50 nursing students two weeks after the initial administration. High consistency between the two administrations was found, as indicated by the intraclass correlation coefficient (ICC) of 0.933.

### 3.5. Testing the Measurement Errors of the Nursing Student Satisfaction Scale (Italian Version)

The measurement errors of the NSSS (Italian version) were examined by calculating the SEM and the SDC, which are essential indicators of the precision and responsiveness of a measurement instrument. All the SEM values fell within acceptable psychometric thresholds, supporting the reliability of the scale. Specifically, the SEMs were 0.22 for Professional and Social Interaction, 0.17 for Curriculum and Teaching, 0.26 for Environment, and 0.11 for the overall NSSS score. Similarly, the SDC values were adequate. Specifically, the SDCs were 0.47, 0.73, and 0.31 for the Professional and Social Interaction, Curriculum and Teaching, and Environment factors, respectively, and 0.62 for the overall NSSS.

## 4. Discussion

The aim of this study was to test the psychometric properties of the NSSS, an instrument designed to measure nursing student satisfaction, by examining its structural validity, construct validity, and internal consistency reliability. To the best of our knowledge, this is the first study to test the factor structure of the NSSS using a confirmatory statistical approach. The findings indicate that the NSSS exhibits strong validity, confirming the multidimensional nature of nursing student satisfaction, which comprises the factors Professional and Social Interaction, Curriculum and Teaching, and Environment, in line with the theoretical model originally proposed by Chen et al. [4]. These results are particularly significant as they demonstrate the stability of the factorial structure following the translation and cultural adaptation process, thereby reinforcing the cross-cultural validity and conceptual coherence of the NSSS. Furthermore, we found evidence of a second-order factorial structure, further supporting the theoretical robustness of the instrument and its use of a total score, which serves as a concise yet comprehensive indicator for evaluating nursing student satisfaction in nursing education contexts.

As for structural validity, we found an excessive residual variance between four pairs of items: #15 and #7, #18 and #17, #26 and #12, and #28 and #9. This suggests that nursing students consider these elements to be related. In particular, #15 (Equipment in the nursing lab was in a good state of repair) and #7 (Sufficient equipment in the nursing lab), as well as #26 (Classrooms had ample space) and #12 (Classroom environment was comfortable), exhibit semantic redundancy, as their wording is very similar, particularly due to the repetition of terms such as “equipment” and “laboratory”. Items #28 (Prepared me to use the nursing process in my clinical practice) and #9 (Prepared me to become a professional nurse) exhibit a conceptual overlap, as the nursing process is widely recognized as a key competency of the nursing profession. Consequently, effective preparation in applying the nursing process is inherently linked to the development of the skills required to assume the professional role of a nurse. Finally, the covariance between item #17 (Faculty encouraged my learning) and item #18 (Program enhanced my problem solving skills) can be explained by the relationship between instructional support and skill development [43]. Students who perceive that they have greater support from faculty are more likely to develop higher-order cognitive skills, thereby explaining the covariance observed. In addition, the “proximity effect” [44], which refers to the influence that the distance between items exerts on participants’ responses toa questionnaire, may have increased the shared variance between the two items. While residual correlations typically suggest potential redundancy and could indicate a need for item reconsideration or removal, this was not deemed necessary in the present study. Indeed, all the items in these pairs demonstrated strong standardized factor loadings, ranging from 0.590 to 0.673, even after correlated residuals were considered, clearly indicating that each item serves as an important and distinct indicator of the underlying construct. Consequently, removing these items could adversely affect the accuracy of construct measurement for the student population studied and is therefore not recommended.

As for the construct validity of the NSSS, several hypotheses were tested. Positive correlations were observed between the three factors and the overall scale, aligning with the theoretical framework proposed by Chen. Furthermore, the results also confirmed our hypothesis that the ANSE would be positively associated with each factor of the NSSS. This finding aligns with prior theoretical and empirical research. According to Pajares and Schunk [45], self-efficacy plays a fundamental role in enhancing individual well-being and motivation, particularly in academic contexts. Self-efficacy traits are known to influence the degree of stress and anxiety experienced while engaging in tasks [46], which may also apply to students’ engagement in academic programs. These results reinforce the relevance of self-efficacy as a determinant of perceived satisfaction in nursing education and confirm the convergent validity of the NSSS.

Our findings confirm that the NSSS demonstrates good reliability in terms of internal consistency, as measured by indices that account for both the factorial structure and multidimensionality of the scales, specifically the composite coefficients and the global reliability index for multidimensional scales. To the best of our knowledge, no previous studies have evaluated the reliability of the NSSS using these appropriate reliability coefficients. These findings confirm that the NSSS is psychometrically sound, ensuring that it accurately and consistently captures the satisfaction of nursing students. In addition, our findings demonstrate that the NSSS possesses strong test–retest reliability, making it a reliable instrument for repeated use in research and educational applications. Finally, the SEM and SDC values of the NSSS indicate an acceptable level of measurement error and that the instrument is capable of detecting significant changes.

These findings support the applicability of this scale in both research and educational contexts for assessing and monitoring nursing student satisfaction. Moreover, considering that student satisfaction may also reflect perceptions of professional identity formation and career development support, the use of this measure could offer valuable insights into how educational experiences influence students’ readiness for future nursing roles. This is in line with the findings of Kajander-Unkuri et al. [47], who showed that satisfaction with nursing education among recent graduates is positively associated with job satisfaction and retention in the nursing profession.

### Limitations and Strengths

Several limitations should be considered when interpreting these findings. First, the use of a convenient and balanced sample obtained through multicenter enrollment resulted in a predominance of female participants, which may have affected the generalizability of the results. Further research involving larger, more diverse, and more representative samples is necessary to adequately evaluate psychometric properties, taking into account the participants’ sociodemographic characteristics. Secondly, the students were asked to rate their satisfaction; thus, the findings may have been affected by recall bias. A key strength of this study was our application of robust analyses and instruments for assessing construct validity.

## 5. Conclusions

The NSSS exhibited robust measurement properties, confirming its validity and reliability as an instrument for assessing nursing student satisfaction within the Italian context. Furthermore, our results indicate that after the translation and cultural adaptation of the scale, the construct of nursing student satisfaction remained consistent with the theoretical model. The NSSS could be valuable for both educational practice and research.

Finally, the NSSS could be useful for screening nursing students with low levels of satisfaction with the curriculum, who may fail to complete their academic duties [18,19]. This screening could support the design of personalized interventions in specific satisfaction domains to support students in their learning.

## Figures and Tables

**Figure 1 nursrep-15-00161-f001:**
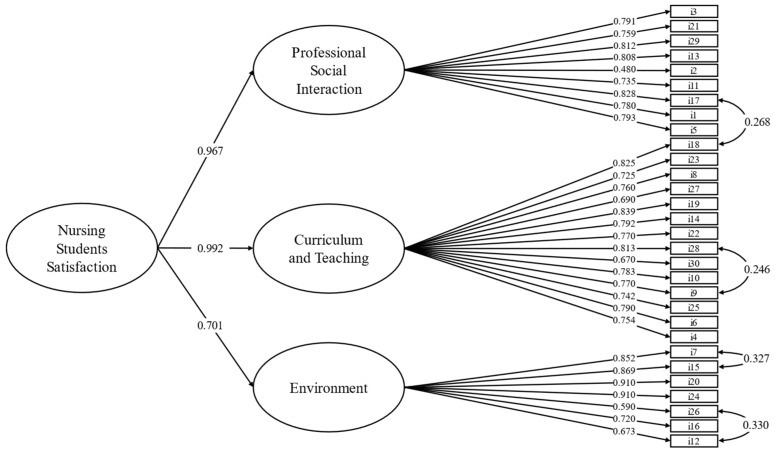
Graphical representation of the confirmatory factor analysis of the Nursing Student Satisfaction Scale (N = 721).

**Table 1 nursrep-15-00161-t001:** Socio-demographic characteristics of the undergraduate nursing students (N = 724).

Variable	N (%)
Sex	
Female	536 (74.0)
Male	188 (26.0)
Year of the course attended	
First	262 (36.2)
Second	248 (34.3)
Third	214 (29.6)
Marital status	
Single	595 (82.2)
Married	122 (16.9)
Divorced	7 (1.0)
Have Children	
No	688 (95.0)
Yes	36 (5.0)
Housing situation	
Alone	44 (6.1)
Living with other students	406 (56.1)
Living with family members	274 (37.8)
Working condition	
Unemployed	632 (87.3)
Employed	92 (12.7)
Nursing was the first choice of degree program	
No	232 (32.0)
Yes	492 (68.0)
	Mean (±SD)
Age	23.56 (5.66)

Legend. SD, standard deviation.

**Table 2 nursrep-15-00161-t002:** Description of the items of the Italian version of the Nursing Student Satisfaction Scale (n = 724).

Items	M	SD	Skewness	Kurtosis
1. I Had positive professional interactions with my faculty	3.41	0.96	−0.54	0.15
2. I Was respected by the nursing staff in the clinical setting	3.64	1.00	−0.63	−0.01
3. Faculty was a positive role model of professional nursing	3.47	1.00	−0.52	−0.09
4. The Program progressed logically from simple to complex concepts	3.46	0.99	−0.55	−0.04
5. I Felt trusted by my nursing faculty	3.57	0.97	−0.60	0.15
6. Syllabus clearly described what was expected by me	3.36	1.05	−0.38	−0.44
7. Sufficient equipment in the nursing lab	2.97	1.27	−0.09	−1.00
8. Faculties collaboratively worked each other	3.18	1.04	−0.27	−0.30
9. Faculty prepared me to become a professional nurse	3.66	0.97	−0.75	0.38
10. Program prepared me to take the professional habilitation	3.39	1.04	−0.56	−0.08
11. I Felt comfortable asking questions of faculty	3.42	1.11	−0.54	−0.36
12. Classroom environment was comfortable	2.80	1.25	−0.01	−1.08
13. I was respected by the faculty	3.45	1.04	−0.53	−0.18
14. Faculty makes their topics interesting	3.36	0.95	−0.47	0.07
15. Equipment in the nursing lab was in good repair	3.14	1.21	−0.29	−0.76
16. Library resources were adequate for my learning needs	2.64	1.22	0.13	−0.87
17. Faculty encouraged my learning	3.15	1.07	−0.32	−0.42
18. Program enhanced my problem solving skills	3.34	1.00	−0.34	−0.26
19. Faculty effectively explained essential concepts	3.55	0.94	−0.67	0.30
20. Nursing lab had ample space	2.89	1.25	−0.11	−0.97
21. Faculty were fair/unbiased in their assessment of my learning	3.06	1.15	−0.29	−0.71
22. Program helped me improve my communication skills	3.45	1.00	−0.62	0.09
23. Faculty were knowledgeable	3.91	0.86	−0.84	1.11
24. Equipment in the nursing lab was up to date	3.01	1.23	−0.16	−0.84
25. Program was relevant to current nursing practice	3.60	0.95	−0.68	0.31
26. Classrooms had a wide space	3.19	1.14	−0.39	−0.63
27. Faculty effectively used technology to enhance my learning	3.21	1.01	−0.32	−0.26
28. I was prepared to use the nursing process in my clinical practice	3.52	0.98	−0.67	0.29
29. Faculty had reasonable expectations of my performance	3.28	0.98	−0.45	0.18
30. I felt confident about my ability to practice in clinical	3.55	1.00	−0.58	0.00

Legend. M, mean; SD, standard deviation.

**Table 3 nursrep-15-00161-t003:** Results regarding the construct validity of the Nursing Student Satisfaction Scale (N = 724 nursing students).

Variable	1	2	3	4
1. Professional and Social Interaction	-			
2. Curriculum and Teaching	0.902			
3. Environment	0.572	0.602		
4. NSSS	0.946	0.969	0.726	
5. ANSEs	0.355	0.346	0.217	0.347

Legend. NSSS, Nursing Student Satisfaction Scale; ANSEs, Academic Nursing Self-efficacy scale. Note. All correlations are significant at the 0.01 level (two-tailed).

**Table 4 nursrep-15-00161-t004:** Internal consistency reliability of the Nursing Student Satisfaction Scale.

	Composite Reliability Coefficient	Global Reliability Index for Multidimensional Scales
NSSS		0.974
Professional and Social Interaction	0.924	
Curriculum and Teaching	0.951	
Environment	0.926	

Legend. NSSS, Nursing Student Satisfaction Scale.

## Data Availability

The dataset is available upon requesting it from the authors.

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
