# Peer review of "Nursing Student Satisfaction Scale: Evaluation of Measurement Properties in Nursing Degree Programs"

_nursrep, 2025, doi:10.3390/nursrep15050161_

Round 1
Reviewer 1 Report
Comments and Suggestions for Authors
Abstract: Line 21- clarify in the first sentence if academic satisfaction is referring to student or faculty satisfaction. Suggest possible revision to "Academic satisfaction in nursing students..." or "Nursing student academic satisfaction". Also suggest specifying in the abstract if this study measures satisfaction in graduate or undergraduate or both.
Line 25 - This sentence is a little confusing. At first read - it sounds like the scale was investigator developed, but then it is stated that it has been tested in American nursing students. Suggestion for rewriting this sentence: The Nursing Students Satisfaction Scale (NS-SS) measures three theoretical domains: Professional Social Interaction, Curriculum and Teaching, and Learning Environment. Although the NS-SS has demonstrated reliability and validity in American nursing students, the psychometric properties have not been tested in a population of Italian nursing students. Therefore, this study aimed to determine the reliability and validity of the NS-SS in Italian nursing students.
Methods: Suggest revising methods to reflect the methodology used to determine reliability and validity of the NS-SS - for example test- retest, interrater reliability, content validity etc. The overall study design could be stated as observational or possibly cross-sectional, but cross-sectional suggests a type of intervention study. My suggestion would be observational. Only include methodology information in this section of the abstract. If there is room in the word count, add a sentence of how the scales were distributed to the nursing students and how many sites were included.
Remove the number of students from the Methods and the sentences following the first sentence. This information should be presented in the results section. Present specifics in the results section.
Suggest revising the abstract conclusion for a stronger statement on the importance of the study.
Throughout the manuscript revise for paragraphs to include at least 3 sentences. Clarify Student academic satisfaction.
Line 55 - letter is missing at end of word
Add what year the tool was developed.
Introduction needs a lot of revision for clarity and organization. Are there other tools that measure student satisfaction? Include a description of reliability and validity testing and why each was chosen for this study.
Methods - specify what type of nursing programs - undergraduate or graduate, and if all years of the program were approached to participate.
How were the participants identified and approached for participation in the study? Was there written informed consent? Who consented the students? Are the researchers working at the university, and if so, was there any conflict of interest?
Move the Cronbach's alpha to the results since this is a reliability study, unless you are reporting previous researchers results, in which case cite this.
Line 141 - the ANSE was not described earlier. There needs to be much more discussion and background/signification regarding the statistical testing and methodology utilized for psychometric testing. This tool should be introduced earlier in the paper, along with presenting information on the reliability and validity testing that was chosen and why each test was chosen.
Line 160 - why were demographic data collected? Include a brief one or two sentence explanation
Line 118 specify which semester, and why this time was chosen.
Describe the 'interviews' conducted by the research assistants. Describe the qualifications of the research assistants - are they students also?
Descriptions of the tests used for reliability and validity and explanations of why each was chosen should be included in the background section.
Reviewer 2 Report
Comments and Suggestions for Authors
The manuscript presents a structured analysis of the use of the tool to assess student satisfaction. In view of the changing environment and the new generations entering education, it is important to know what aspects need to be evaluated in the course of education. The publication contains references to the evaluation analyses of the tool. The issues related to the variables are comprehensively described. The methodology is sufficiently described and the statistical analyses are not in doubt.
Author Response
Comments 1: The manuscript presents a structured analysis of the use of the tool to assess student satisfaction. In view of the changing environment and the new generations entering education, it is important to know what aspects need to be evaluated in the course of education. The publication contains references to the evaluation analyses of the tool. The issues related to the variables are comprehensively described. The methodology is sufficiently described and the statistical analyses are not in doubt.
Response 1: We thank the reviewer for the positive and encouraging feedback. We appreciate your recognition of the clarity of our methodological approach, the comprehensiveness of the variable description, and the appropriateness of the statistical analyses. We are pleased that the relevance of evaluating student satisfaction in the evolving educational context was acknowledged.

Reviewer 3 Report
Comments and Suggestions for Authors
The work presented is technically appropriate. Meets the proposed objectives by applying the scientific methodology used for these purposes in an appropriate and rigorous manner. In this sense, there is adequate use of psychometric techniques, which allows them to support their conclusions correctly.
This study aimed to test the psychometric characteristics (structural validity, construct validity, and reliability of internal consistency) of the NS-SS in a sample of students. The work provides evidence regarding the psychometric properties of this instrument, which aims to provide information regarding the satisfaction of nursing students specifically in Italy. In this sense, its adaptation to Italian and the culture of this country stands out. In this sense, its relevance may be specific to this context, however, as the authors indicate in the document, there is no important information regarding it in Italian.
A correct use of psychometric tools is observed according to verifying the psychometric characteristics of the instrument. Therefore, the results are established with adequate rigor, considering the declared methodological processes.
Considering the above, the conclusions are consistent with the evidence and the results presented. In turn, they allow an adequate answer to the main question posed.
Presents adequate references to justify their decisions and support the essential aspects of the theory presented. However, it is important to note that there are few references from the last five years. Similarly, along with updating their references, it is proposed that the authors provide more information regarding studies related to the instrument, even if this corresponds to characterizations made in other languages. This will allow for a better appreciation of the structure and reliability of the instrument, and of the debates that may exist regarding these aspects, which, in turn, will strengthen the conclusions of the study.
Reviewer 4 Report
Comments and Suggestions for Authors
Comment to the nursrep-3549181-peer-review-v1
- The definition of academic satisfaction in Nursing Degree Programs.
- The name of the Nursing Student Satisfaction Scale does not adequately capture academic satisfaction. Some references use student academic satisfaction to define this concept.
- This research needs to provide academic satisfaction models or theories to support the measurement tool of academic satisfaction.
- Academic satisfaction in Nursing Degree Programs may include the concept of nursing career development.
- Using structural validity, construct validity, and internal consistency reliability to test the psychometric characteristics of the NS-SS was adequate.
- There are four pairs of items, #15 and #7, #18 and #17, #26 and #12, and #28 and #9. This suggests that nursing students consider these elements to be relevant. Therefore, it is recommended to discuss whether some highly relevant items need to be streamlined or deleted.
- The research methods, research tools, and research results of this study are properly described.
- However, there is a lack of discussion on other theories and concepts related to student academic satisfaction, which may affect the expansion of the measurement scope of this concept or the failure to increase innovative results.

Round 2
Reviewer 1 Report
Comments and Suggestions for Authors
Thank you for providing the revised document. There are still changes to the introduction that would make the article much stronger and impactful for the reader.
It still appears that there are paragraphs with 2 sentences.
Change NS-SS to NSSS as it appears in other literature
I'm not sure if the Polit & Beck references is appropriate for this article
If Line 59 is the definition of student satisfaction, change the sentence to 'Student satisfaction can be defined as a subjective assessment by the student, which is determined by the gap between the student's expectations and their perception of the educational experiences they have received."
Introduction Line 59 - suggest making Students satisfaction singular - "Student satisfaction..." Throughout the manuscript suggest careful revision of students and/or student, changing students' satisfaction to singular to read 'student satisfaction' or 'nursing student satisfaction'.
Line 66, Change to ' According to Chen et al, student satisfaction is the result of......
Line 71 -82 - This section would be improved by summarizing evidence that shows high levels of perceived satisfaction improve retention, etc. If it summarizes evidence from previous studies, suggest something similar to the following revision: Quality indicators in undergraduate nursing education programs include XXXX. Therefore, student satisfaction is an important indicator for faculty to measure.
It sounds like you are saying that evidence demonstrates that universities reporting higher perceived student satisfaction experience improved undergraduate nursing student retention, lower academic costs, and an improved reputation. Briefly summarize each article with 2-3 sentences (it looks like from the reference page 10, 13, and 14) regarding the link between student satisfaction and undergraduate nursing student retention, student satisfaction and lower academic costs - also explain what is meant by lower academic costs, and the link between student satisfaction and improved reputation - and the importance of each of these outcomes. Without reading the literature, I would assume that improved retention rates are important because they can lead to higher numbers of licensed nurses, which improves the workforce. I am not sure about the academic costs, but an improved reputation is likely linked to better collaboration opportunities, accreditation, practice partnerships, etc. What the reader needs to see here is:
What is student satisfaction?
Why is it important - indicators of quality education, student satisfaction is a part of this, evidence shows student satisfaction impacts X and that is important because X.
How do we measure student satisfaction?
Line 88 change The nursing degrees program to: Nursing degree programs differ from other disciplines because they require theoretical knowledge linked to practical training, critical thinking, and clinical judgement skills, along with the ability to communicate effectively with patients and caregivers. Therefore, nursing degree programs may experience difficulties with student retention.' Suggest moving these 2 sentences to the section where you summarize literature that links student satisfaction to retention.
Line 92: Change to "Instruments for measuring nursing student satisfaction....
Line 100 revise to:.....Rossini et al concluded that the best available instrument for measuring nursing student satisfaction was the Nursing Student Satisfaction Scale
Line 375 Explain what the proximity effect is
